# Immune and Oxidative Response against Sonicated Antigen of *Mycoplasma capricolum subspecies capripneumonia*—A Causative Agent of Contagious Caprine Pleuropneumonia

**DOI:** 10.3390/microorganisms10081634

**Published:** 2022-08-12

**Authors:** Rather Izhar Ul Haq, Oveas Rafiq Parray, Qurat Ul Ain Nazir, Riyaz Ahmed Bhat, Showkat Ahmad Shah, Majid Shafi Kawoosa, Ali A. Rabaan, Mohammed Aljeldah, Basim R. Al Shammari, Mohammed S. Almogbel, Nada Alharbi, Reem Alrashoudi, Amal A. Sabour, Rana A. Alaeq, Maha A. Alshiekheid, Saleh A. Alshamrani, Aqel Albutti, Ameen S.S. Alwashmi, Kuldeep Dhama, Mohd. Iqbal Yatoo

**Affiliations:** 1Mycoplasma Laboratory, Faculty of Veterinary Sciences and Animal Husbandry, Shuhama, Alusteng, Srinagar 190006, Jammu and Kashmir, India; 2Molecular Diagnostic Laboratory, Johns Hopkins Aramco Healthcare, Dhahran 31311, Saudi Arabia; 3College of Medicine, Alfaisal University, Riyadh 11533, Saudi Arabia; 4Department of Public Health and Nutrition, The University of Haripur, Haripur 22610, Pakistan; 5Department of Clinical Laboratory Sciences, College of Applied Medical Sciences, University of Hafr Al Batin, Hafr Al Batin 39831, Saudi Arabia; 6Department of Medical Laboratory Sciences, College of Applied Medical Sciences, University of Hail, Hail 4030, Saudi Arabia; 7Department of Basic Medical Sciences, Unaizah College of Medicine and Medical Sciences, Qassim University, Buraydah 51452, Saudi Arabia; 8Department of Clinical Laboratory Sciences, College of Applied Medical Sciences, King Saud University, Riyadh 11461, Saudi Arabia; 9Department of Botany and Microbiology, College of Science, King Saud University, Riyadh 11451, Saudi Arabia; 10Department of Medical Laboratories Technology, Faculty of Applied Medical Science, Taibah University, Al Madinah Al Munawarh 42353, Saudi Arabia; 11Department of Clinical Laboratory Sciences, College of Applied Medical Sciences, Najran University, Najran 61441, Saudi Arabia; 12Department of Medical Biotechnology, College of Applied Medical Sciences, Qassim University, Buraydah 51452, Saudi Arabia; 13Department of Medical Laboratories, College of Applied Medical Sciences, Qassim University, Buraydah 51452, Saudi Arabia; 14Division of Pathology, ICAR-Indian Veterinary Research Institute, Izzatnagar, Bareilly 243122, Uttar Pradesh, India

**Keywords:** antigen, contagious caprine pleuropneumonia, immunity, mycoplasma, oxidative stress, rabbit, vaccine

## Abstract

Vaccines are vital for prevention and control of mycoplasma diseases. The exploration of a vaccine candidate for the development of a vaccine is imperative. The present study envisages the evaluation of immune and oxidative response against an adjuvanted, sonicated antigen of *Mycoplasma capricolum subsp. capripneumonia* in male Angora rabbits (1 year old, 2 kg) divided in four groups, each having six animals. Group 1 was the healthy control and received 1 mL PBS via subcutaneous route. Group 2 was administered 1 mL of saponin-adjuvanted and -sonicated antigen, Group 3 was given 1 mL of montanide ISA 50-adjuvanted and-sonicated antigen, and Group 4 was given 1 mL of standard vaccine via subcutaneous route. Animals were evaluated for cellular and humoral immune response and oxidative parameters at 0, 7, 14, 21, and 28 days of the study. Total leukocytic, neutrophilic, and basophilic counts showed a significant (*p* < 0.05) increase in vaccinated groups compared to the healthy group on most of the intervals. TNF-α levels were significantly (*p* < 0.05) higher in the Group 2 than the Group 1 at all the time intervals and more comparable to Group 4 than Group 3. IL-10 levels were significantly (*p* < 0.05) higher in vaccinated groups compared to the healthy group on days 14, 21, and 28, but were lower in Group 3 than in Group 2 and Group 4. More hypersensitivity as inflammation and histopathological cellular infiltration in the ear was produced in Group 2 and Group 4 than in Group 3. IgG levels were significantly (*p* < 0.05) higher in Group 2 and Group 4 than in Group 3 on days 14 and 21. Antibody titers were comparatively higher in Group 4, followed by Group 2 and 3, than Group 1. Significantly (*p* < 0.05) higher oxidant and lower antioxidant values were noted in Group 2 and 4 compared to Group 3 and Group 1 on most of the intervals. The TLC and antibody titer showed increasing trend throughout the trial, whereas TNF-α, IgG, L, M and E started decreasing from day 14, and IL-10, N and B started decreasing from day 21. This study concludes that the saponin-adjuvanted and-sonicated antigen induces comparatively higher immune response than montanide but is associated with oxidative and inflammatory reactions.

## 1. Introduction

Mycoplasmosis, a disease syndrome caused by different mycoplasma organisms [1,2], is one of the challenging and continuous threats to small ruminant farming resulting in heavy morbidity (upto 100%), mortality (10–100%) and massive economic loss [1,3,4]. These smallest wall-less microbes are resistant to common antibiotics affecting the cell wall [5], and the trend of frequent and prolonged use of antibiotics may lead to resistance against other antibiotic classes [6]. Hence, the development of vaccines against mycoplasma is imperative.

Of the mycoplasma diseases, contagious caprine pleuropneumonia (CCPP) caused by *Mycoplama capricolum* subspecies *capripneumonia* (Mccp) is a highly contagious, severe and fatal disease of small ruminants, especially goats, and is spreading from endemic to non endemic areas; hence, it is emerging as a novel transboundary threat to goat-farming countries [7,8,9,10,11]. Its morbidity is as high as 100% and its mortality is around 80% [7,9,12]. This disease is manifested by extreme serofibrinous pleuropneumonia resulting in respiratory distress, coughing, nasal discharge, pleurodynia, fluid exudation, fever, anorexia and death [7,9]. Various antibiotics minimize severity and offer temporary relief; however, the extended use of antibiotics may result in detrimental effects including antibiotic-specific side effects and the risk of antibiotic resistance [9,13]. Besides, treated animals usually remain as carriers for the infection for quite some time; hence, they pose risks of further disease spread [2,13,14,15]. Hence, prevention by vaccines is an imperative and permanent solution [2,15,16].

The Office International des Epizooties (OIE) recommends a saponin-inactivated and -adjuvanted whole Mccp cell-based vaccine against CCPP. It provides considerable immunity; however, the use of saponin may have some limitations and undesirable effects; hence, the exploration of novel adjuvants is important [17]. Immune response has important role in pulmonary infections and their protection including mycoplasma pneumonia; however, the involvement of cellular and humoral immune response needs to be explored [15,16,17]. Further oxidative stress is involved in the course of pulmonary infection; however, oxidative response against mycoplasma vaccines is rarely evaluated [15,16,17]. Hence, the present study is aimed at exploring a sonicated antigen as vaccine candidate from locally isolated Mccp and the evaluation of immune and oxidative response against different adjuvants.

## 2. Materials and Methods

### 2.1. Preparation and Adjuvanation of Mycoplasma Antigen 

The antigen was isolated from the broth culture of Mccp maintained at our Mycoplasma Laboratory by the OIE protocol followed by sonication [10,13,18]. Briefly, Mccp broth culture (20 mL) was centrifuged at ≥12,000 rpm for 20 min. Pellet was suspended in desired amount of PBS, repelleted and further suspended. This was followed by sonication (JY92-IIN Ultrasonic Homogeniser, Helix Bioscience, Delhi, India). The protein concentration of antigen was estimated by Bicinchoninic Acid Assay kit (BCA). A dose of 0.15 mg of antigen was adjuvanted by 3 mg of saponin in 1 mL of PBS, making one dose of vaccine [19]. Similarly, 0.15 mg of montanide ISA 50 was used for adjuvanating 0.15 mg of the protein antigen in 1 mL of PBS and used as one dose of vaccine [17].

### 2.2. Experimental Trial

Twenty-four pre-adapted and pre-screened disease free experimental male Angora rabbits (1 year age, 2 kg) were divided into four groups, with each group containing six animals. Group 1 was the healthy or negative control. Group 2 was used as the test group and was given 1 mL of saponin-adjuvanted antigen subcutaneously. Group 3 was given 1 mL of montanide ISA 50-adjuvanted antigen subcutaneously. Group 4 was given standard antigen/vaccine and served as standard group. Blood samples were collected at days 0, 7, 14, 21 and 28 of the trial. About 3 mL of blood from heart using 24 G needle was taken in heparin vial for hematology estimations, and 3 mL of blood was taken in clot activator vial for harvesting serum for evaluating immune and oxidative indices.

### 2.3. Evaluation of Immune Indices

For cellular immunity (CI), parameters including total leukocyte count (TLC) and differential leukocyte count (DLC) were evaluated [20], as well as proinflammatory cytokine (TNF-α) and anti-inflammatory cytokine (IL-10) by using ELISA kits (Sincere Biotech Co., Ltd., New Taipei, China), and the delayed type of hypersensitivity (DTH) was evaluated by measuring inflammatory changes in the ear and histopathological infiltration [21] with slight modification, followed by histopathology [22]. For humoral immunity (HI), immunoglobulins IgG was evaluated by ELISA kit (Sincere Biotech Co., Ltd., New Taipei, China). Antibody titer was measured by slide agglutination test as described by Parray et al. [10].

### 2.4. Evaluation of Oxidative Indices

Oxidative status was evaluated by estimating Total Oxidant Status (TOS) and Total Antioxidant Status (TAS) at weekly intervals from day 0 to day 28 [23,24].

### 2.5. Hematological Evaluation

Hematological evaluation was done by measuring total erythrocyte count (TEC) as per the standard conventional procedure [20] and hemoglobulin (Hb) was measured as per Sahli’s acid haematocrit method [25].

### 2.6. Statistical Analysis

Repeated measure ANOVA was used for determining mean and standard error for various parameters at a significance of *p* < 0.05. All the statistical tests were carried by SPSS software version 20 (IBM Corp., Armonk, NY, USA). 

## 3. Results

### 3.1. Effect on Immune Response

#### 3.1.1. Cellular Immune Response

##### Effect of Adjuvanted, Sonicated Antigens on Total Leucocyte Count (TLC) and Differential Leucocyte Count (DLC)

The effect of adjuvanted, sonicated antigens on TLC, N, L, M, E and B in rabbits is given in Figure 1a–f. Between groups, no significant (*p* ≥ 0.05) differences were noted on day 0 of the trial. TLC differed significantly (*p* < 0.05) in vaccinated groups on most of the intervals; however, values were higher in Group 2 and 4 compared to Group 3. Neutrophil counts differed significantly (*p* < 0.05) in the vaccinated Group 4 on most of the intervals; however, Group 2 and 3 did not differ significantly (*p* ≥ 0.05). Significantly higher lymphocyte counts were noted in Group 4 compared to Group 2 and 3 from day 14 to 28. No significant (*p* ≥ 0.05) difference in lymphocyte counts was noted between various groups on various times. Non-significantly (*p* ≥ 0.05) higher monocyte counts were noted in Group 2 and 3 compared to Group 3 on day 7 and 14. A significant (*p* < 0.05) increase in monocyte counts in Group 4 on day 7 was noted. Eosinophill counts did not differ significantly (*p* ≥ 0.05) in vaccinated groups on all days. Basophill counts differed significantly (*p* < 0.05) between Group 2 and Group 3 on day 28. Other groups did not differ significantly (*p* ≥ 0.05).

Within groups, no significant change (*p* ≥ 0.05) in TLC in Group 1 was noted on various time points. In vaccinated Groups 2, 3 and 4, TLC increased significantly (*p* < 0.05) upto day 21, thereafter showing no significant (*p* ≥ 0.05) change. In vaccinated Group 2, neutrophil counts increased significantly upto day 7, and in Group 3 and 4 upto day 14, and thereafter, the non-significant (*p* ≥ 0.05) decrease in neutrophil counts was noticed. On day 7, lymphocyte counts were significantly (*p* < 0.05) higher in vaccinated groups compared to day 0. From day 14, lymphocyte counts showed non-significant (*p* ≥ 0.05) fall in all vaccinated groups. A non-significant (*p* ≥ 0.05) increase in monocyte counts was noted in all test groups on day 14, and thereafter, a non-significant (*p* ≥ 0.05) decrease was noted. A significant (*p* < 0.05) increase in eosinophill counts in all test groups on day 7 was noted. A significant (*p* < 0.05) decrease in eosinophill counts in Group 2 from day 7 to day 21 wasnoted. A non-significant (*p* ≥ 0.05) decrease in eosinophill counts was noted in Group 4 at all times and in Group 3 from day 14 to 21. A significant (*p* < 0.05) increase in basophill counts in all test groups upto day 14 was noted. In Group 2, there was significant (*p* < 0.05) increase in basophill counts upto day 14. A non-significant (*p* ≥ 0.05) decrease in basophill counts was noted in all groups from day 21 to 28.

##### Effect of Adjuvanted, Sonicated Antigens on Proinflammatory Cytokines (TNF-α)

Effect of adjuvanted, sonicated antigens on TNF-α in rabbits is given in Figure 2. Between the groups, no significant (*p* ≥ 0.05) difference in TNF-α levels was noted in various groups on day 0 of the trial. TNF-α levels in Group 4 were significantly higher followed by Group 3 and Group 2 compared to healthy Group 1 on day 14. Levels of TNF-α were non-significantly (*p* ≥ 0.05) higher in Group 4, followed by Groups 2 and 3, on days 21 to 28.

Within the groups, TNF-α levels showed no significant (*p* ≥ 0.05) difference in Group 1 over various time periods. In Groups 2, 3 and 4, TNF-α levels increased significantly (*p* < 0.05) upto day 14 and then decreased non-significantly (*p* ≥ 0.05). Levels of TNF-α decreased non-significantly (*p* ≥ 0.05) in various groups from day 21 to 28.

##### Effect of Adjuvanted, Sonicated Antigens on Anti-Inflammatory Cytokines (IL-10)

The effect of adjuvanted, sonicated antigens on IL-10 in rabbits is given in Figure 3. Between groups, no significant (*p* ≥ 0.05) differences in IL-10 levels were noted in various groups on day 0 of trial. IL-10 levels were significantly (*p* < 0.05) higher in vaccinated groups compared to Group 1 on days 14, 21 and 28, but were lower in Group 2 than Group 1 and 4 on days 14 and 21.

Significantly (*p* < 0.05) higher IL-10 levels were noted in vaccinated groups on day 14 and 21 compared to day 0.

##### Effect of Adjuvanted, Sonicated Antigens on Delayed Type of Hypersensitivity (Ear Thickness)

Effect of adjuvanted, sonicated antigens on ear thickness in rabbits is given in Figure 4a–c. A significant (*p* < 0.05) increase in ear thickness was noticed in Group 2 followed by Group 4 and Group 3 at all times from day 7, except for a non-significant (*p* ≥ 0.05) difference between Group 2 and Group 3 on day 28.

Ear thickness increased significantly (*p* < 0.05) in all groups from day 0 to day 14 followed by significant (*p* < 0.05) decrease from day 14 to day 28 except for a non-significant (*p* ≥ 0.05) decrease in Group 3 from day 21 to day 28. 

### 3.2. Effect on Humoral Immune Response

#### 3.2.1. Effect of Adjuvanted, Sonicated Antigens on IgG Levels 

Effect of adjuvanted, sonicated antigens on IgG in rabbits is given in Figure 5. Between groups, no significant (*p* ≥ 0.05) differences in IgG levels were noted in various groups on day 0 of the trial. IgG levels were significantly (*p* < 0.05) higher in Group 2 and Group 4 than Group 3 on days 7, 14 and 21.

Within groups, IgG levels increased significantly (*p* < 0.05) in vaccinated groups upto day 14 and thereafter decreased significantly (*p* < 0.05) upto day 21, and in Group 2 and Group 4 upto day 28, and the levels increased non-significantly (*p* ≥ 0.05) in Group 2.

#### 3.2.2. Effect of Adjuvanted, Sonicated Antigens on Antibody Titer 

Effect of adjuvanted, sonicated antigens on antibody titer in rabbits is given in Figure 6. Significantly (*p* < 0.05) higher antibody titers were noted in Group 4 followed by Group 2 and Group 3 at all times from day 7 except for a non-significant (*p* ≥ 0.05) difference between Group 2 and Group 3 on days 7 and 14 and between Group 3 and Group 4 on day 28.

Antibody titers increased significantly (*p* < 0.05) in all vaccinated groups from day 0 to day 28.

#### 3.2.3. Effect of Adjuvanted, Sonicated Antigens on Oxidative Indices

##### Effect of Adjuvanted, Sonicated Antigens on Total Oxidative Status (TOS)

The effect of adjuvanted, sonicated antigens on total oxidative status in rabbits is given in Table 1. Vaccinated groups differed significantly (*p* < 0.05) in TOC values at days 14 and 21, with Group 2 having higher values followed by Group 4 and Group 2.

TOS values differed significantly (*p* < 0.05) between different time intervals in all test groups with increase in values upto day 7, followed by a decreasing trend afterwards, except for a non-significant (*p* ≥ 0.05) decrease in Group 3 and 4 on day 28.

##### Effect of Adjuvanted, Sonicated Antigens on Total Antioxidant Status (TAS) 

The effect of adjuvanted, sonicated antigens on the total antioxidant status in rabbits is given in Table 2. Non-significantly (*p* ≥ 0.05) higher TAS levels were noted in Group 3 compared to Group 2 and 4 at various intervals.

Significantly (*p* < 0.05) lower TAS values were noted in Group 2 and Group 4 on day 7 compared to day 0. TAS values increased non-significantly (*p* ≥ 0.05) in all groups from days 7 to 28, except for a significant increase in Group 4 on day 28 compared to days 7, 14 and 21.

### 3.3. Effect on Hematological Parameters

#### 3.3.1. Effect of Adjuvanted, Sonicated Antigens on Total Erythrocyte Count (TEC) 

The effect of adjuvanted, sonicated antigens on total erythrocytes count in rabbits is given in Table 3. TEC values were significantly higher in Group 3 than Groups 2 and 4 on day 7. Non-significantly (*p* ≥ 0.05) higher TEC values were noted in Group 3 compared to Groups 2 and 4 most of the other times.

Significantly (*p* < 0.05) lower TEC values were noted in Group 2, Group 3 and Group 4 on day 7 compared to day 0. On day 14, Group 2 and Group 4 showed a significant (*p* < 0.05) increase in TEC values. On day 21, all groups showed significantly (*p* < 0.05) higher TEC values compared to day 7 and day 14. On day 28, Group 2 and Group 4 showed significantly (*p* < 0.05) higher TEC values compared to other days.

#### 3.3.2. Effect of Adjuvanted, Sonicated Antigens on Hemoglobin (Hb)

Effect of adjuvanted, sonicated antigens on hemoglobin (Hb) count in rabbits is given in Table 4. Non-significantly (*p* ≥ 0.05) higher Hb values were noted in Group 3 compared to Groups 2 and 4 at various intervals.

A significant (*p* < 0.05) decrease in Hb values in all test groups was noted on day 7. Compared to day 0, Hb values in Group 2 and Group 4 were significantly (*p* < 0.05) lower than day 28 when the other groups showed no significant (*p* ≥ 0.05) difference between Hb values at different intervals.

## 4. Discussion

Vaccination against CCPP has always been focus of discussion whenever livestock development strategies are being discussed, especially in endemic regionslike East Africa, the Middle East, Asia and some European countries [9,15,16,26]. Though some vaccines have been developed against CCPP, including live or inactivated vaccines, each class has its own merits and demerits. Pathogen-based live vaccines can provide longer immunity, reducing the dose requirement and the overall cost; however, they may pose a risk of infection [9,15,26]. The currently used inactivated CCPP vaccines require higher doses for administration, 6-month repetitions, have adjuvantrelated adverse reactions and are costly [9,15]. Most of these vaccines are lacking in countries where they are of the utmost importance. Hence, isolating local strains of Mccp and exploring antigens or vaccine candidates of Mccp is important. The evaluation of novel adjuvants and immune response can help in overcoming some of the limitations associated with saponin. It will provide safe, effective, cheap and readily available vaccines [9,15]. The main aim of this study was to focus on few of these issues.

### 4.1. Effect on Immune Response

The increase in total leukocyte count in vaccinated groups may be due cellular immune response to antigens of the vaccine, resulting in the elevation of white blood cell counts. Cellular immune response against mycoplasma has been reported [27,28]. Higher TLC in CCPP-affected goats has also been reported [13]. Comparatively higher TLC in saponin-adjuvanted vaccinated groups than the montanide-adjuvanted and healthy groups at different time intervals indicates the cellular immune response maintaining over time and remaining constant for quite some period. Saponin is known to induce cellular immune response [29,30]. Higher TLC in CCPP-infected animals indicates a cellular response against mycoplasma antigens [13]. A cellular immune response against mycoplasma antigens has been reported [31,32]. An increase in neutrophil counts in vaccinated groups reflects a neutrophil-mediated cellular response against vaccine antigens. Neutrophil infiltration in an infected group may be due to neutrophil-mediated cellular immune response against mycoplasma infection. Increased neutrophil counts in CCPP-affected goats have been reported [13,33]. Comparatively higher neutrophil counts in vaccinated groups on different time intervals suggests a neutrophil-mediated cellular immune response that remains at a higher level for some period. Higher neutrophil counts in CCPP-affected goats have also been reported [13,33]. Neutrophil-mediated pulmonary infiltration in CCPP-affected goats has been noted [33]. Comparatively higher lymphocyte counts in vaccinated groups indicate a cellular immune response against mycoplasma and vaccines. Abdelsalam et al. reported similar lymphocyte counts in Mccp-infected goats; however, they noted higher lymphocyte counts in healthy control goats than we noted in healthy control rabbits [34]. A non-significant difference in monocyte, eosinophil and basophil counts reflects lesser role of these cells in mycoplasma infections. This is in corroboration with Abdelsalam et al., who also reported a non-significant difference in monocyte and eosinophil counts between healthy and Mccp-infected goats [34].

The cellular immune response may operate through a release of cytokines or free radicals against antigens of mycoplasma [31,32,35]. Thus, an increase in levels of TNF-α and IL-10 in vaccinated groups, especially in saponin-adjuvanted groups, may be due to stimulation of the immune response, resulting in the enhanced production of cytokines. Mycoplasma antigens and saponins induce production of cytokines including TNF-α and IL-10 [30,36,37]. An increase in levels of TNF-α in CCPP-affected goats compared to healthy goats has been noted [6,38]. The proinflammatory cytokine TNF-αis produced by inflammatory cells, especially macrophages, in response to a stimulation like mycoplasma infection [38,39,40]. TNF-α is involved in pathogenesis of mycoplasma infections and can also reflect immune response and protectioninduced by vaccines [38]. Levels of proinflammatory cytokines need to be countered by anti-inflammatory cytokines. Significantly higher IL-10 levels in vaccinated groups, especially in the saponin-adjuvanted group compared to other groups, may indicate the protective response of vaccines against inflammatory mechanisms induced by mycoplasma and saponin. IL-10, being an anti-inflammatory cytokine, helps to counter proinflammatory cytokines and, hence, inflammatory reactions. Saponin induces IL-10 production [41]. The involvement of IL-10 in mycoplasma infections in sheep and goats has been elucidated [42,43], and its anti-inflammatory nature has been reported [43]. IL-17-mediated neutrophilic infiltration as a possible mechanism for lung injury in CCPP-affected goats has been noted [33]. Nevertheless, montanide induced a lesser production of TNF-α and IL-10 compared to saponin. This may be due lesser inflammatory role of montanide [17].

A more inflammatory reaction in the ear due to hypersensitivity reaction in vaccinated groups may be due to cellular infiltration, causing the accumulation of inflammatory mediators and exudates, resulting in an increase in swelling and thickness and higher number of inflammatory cells, as noted on histopathology (Figure 4c). Rabbit ear hypersensitivity [44] and skin hypersensitivity have been evaluated for immune response in rabbits [45].

The activation of the cellular immune response initiates the humoral immune response, leading to the production of antibodies [46,47]. Elevated levels of IgG in vaccinated groups might indicate a humoral immune response against the sonicated antigen in the vaccine. A humoral immune response against mycoplasma vaccine in rabbits has been noted [48]. The initial humoral response against capsular polysaccharide (CPS) of Mccp is through IgM followed by IgG. The IgM response diminishes from day 14, whereas the IgG response remains unchanged or increases as reported in experimental Mccp infections in goats [49]. An increase in levels of IgG upto 6 months followed by constant levels or decreasing levels may suggest a rise of the antibody response against the antigen for upto 6 months, attaining a peak for some time and then decreasing. Immunity may last from 6 months against sonicated Mccp antigens in Freund’s complete and/or incomplete adjuvants or for up to one year against lyophilised F38 mycoplasma antigen adjuvanted with saponin [18,50]. Baziki et al. have also studied seroconversion or antibody response against the Mccp protein antigen and commercial vaccines over a period of 38 days [51].

The higher antibody titer in vaccinated groups indicates the development of humoral immunity due to vaccines in these groups compared to the control group. The humoral immune response is considered to be important for the immunity against Mccp [48,49]. The humoral immune response against the mycoplasma vaccine is determined by sufficient antigens and adjuvants present in the vaccine and the quality of vaccine, which ultimately reflects intensity and duration of antibody titer [52,53]. An appropriate antigen-adjuvant produces an adequate immune response, and thus, a sufficient antibody titer. The higher antibody titer in the group vaccinated with the saponin-adjuvanted and-sonicated antigen might be due to both the availability of more epitopes for immune response in the sonicated antigen containing internal and external constituents of Mccp cells, which were produced after the disruption of Mccp cells by sonication, and the humoral immune response activity of the saponin adjuvants [18,30,54]. The number of vaccinations also determines the antibody titers [55]. The rise of the titer with time is in accordance with Sunder et al. [48], who noted an antibody response against *M. mycoides subsp. mycoides* by about day 7, whereas Rana and Srivastava [56] noted an antibody response against *M. mycoides* subsp. *capri* saponin vaccine by about day 3, continuing to week 14. Srivastava noted an antibody response against *M. capri* at around day 7 that lasted upto week 12 [57]. This difference in antibody response may be due multiple reasons related to the vaccine, adjuvant or animal. Thus, the comparatively better overall immune response noted in Groups 2 and 4 compared to Group 3 may be due to the better adjuvanating nature of saponin compared to montanide ISA 50. However, no adverse reaction was noted in Group 3 in comparison to Groups 2 and 4. This may be due to lesser inflammatory nature of montanide ISA50 compared to saponin resulting in less adverse reactions in Group 3. A better immune response in the saponin-adjuvanted vaccinated groups compared to montanide ISA50-adjuvanted vaccinated groups is in corroboration with Ayelet et al., who also reported a better immune response against the inactivated CCPP vaccine adjuvanted with saponin compared to montanide ISA 50; however, post vaccinal reactions were not noted in the montanide ISA 50-adjuvanted vaccine [17].

### 4.2. Effect on Oxidative Stress

Oxidative stress has role in health and disease of small ruminants [58,59] In infectious diseases, oxidative stress has also been reported [60]. It is involved in respiratory diseases including pneumonia [58,59]. Mycoplasma-induced reactive oxygen species are responsible for pathogenesis of mycoplasmosis diseases in small ruminants [13,61]. Hence, oxidative stress has role in mycoplasma infections.

An increase in TAS levels in vaccinated groups may suggest enhanced antioxidant capacity, whereas a decrease in levels of TOS indicates the minimization of oxidative stress. The increase in levels of TOS and decrease in levels of TAS in the saponin-adjuvanted vaccinated group reflect oxidative stress resulting in the generation of oxidative radicals and consumption of antioxidants. Both mycoplasma and the saponin adjuvant are reported to cause inflammatory and oxidative stress [13,62]. An increase in total oxidant status and decrease in total antioxidant status in oxidative stress conditions have been reported [63,64].

Excessive inflammatory reactions result in oxidative stress [13,65]. Inflammatory cytokines result in oxidative damage. An increase in number of neutrophils results in the production of excessive IL-17-mediated neutrophilic infiltration followed by neutrophilic chemokine secretion and ultimately oxidative stress. Ma et al. noted decreased superoxide dismutase (SOD) and increased nitrous oxide (NO) and malondialdehyde (MDA) in CCPP-affected goats reflecting oxidative stress [33]. Yatoo et al. also reported elevated TOS and decreased TAS in CCPP-affected goats [13]. 

Mycoplasma infection may result in oxidative stress due to generation of ROS and attenuation of cellular antioxidant capacity or a combination of both [66]. It can lead to DNA damage in cells resulting in destruction to antioxidant defense system and, hence, the inability to counter oxidative stress [67]. These mycoplasma-generated reactive oxygen species can also induce apoptosis of cells [68]. This all adds to the pathogenesis of mycoplasmosis.

### 4.3. Effect on Haematological Parameters

The decrease in Hb and TEC in saponin-adjuvanted vaccinated groups compared to the montanide ISA 50-adjuvanted group may be due to the hemolytic nature of saponin, resulting in the reduction of hematological parameters or their severe immune or inflammatory nature, resulting in the alteration of Hb and TEC. Changes in Hb and TEC following vaccination in rabbits have been reported by Stancu et al. [69]. Hemolytic activity of saponins is well known but this does not relate to adjuvant activity [29].

## 5. Conclusions

Vaccines against mycoplasma infections are an important tool to counter these emerging and transboundary threats besides minimizing the menace of antibiotic resistance. However, a lack of vaccines in most of the small ruminant-rearing countries of the world, and the higher cost of vaccines wherever available, makes the usage of alternate therapeutic strategies inevitable. Thus, the exploration of mycoplasma antigens of local isolates and the development of vaccines is imperative for both immune effectiveness and cost effectiveness.

The present study revealed that the saponin-adjuvanted and -sonicated antigen induced a comparatively higher cellular immune response compared to the montanide-adjuvanted and -sonicated antigen in terms of TLC, DLC, TNF-α, IL-10 and DTH.A comparatively higher humoral immune response was noticed in the saponin adjuvant as compared to montanide in terms of IgG and antibody titer. A comparatively higher TOS was noted in the saponin adjuvant than the montanide adjuvant, whereas a higher TAS was noted in montanide as compared to saponin. Lower TEC and Hb values were noted in the saponin-based adjuvanted group as compared to the montanide-based adjuvanted group. This study reflects the better immune-eliciting capability of saponin-adjuvanted and -sonicated antigen compared to montanide, but the former shows more oxidative and inflammatory reactions than the latter, as revealed by elevated TOS, TNF-α and IL-10 and decreased TAS levels. Thus, saponin as an adjuvant induces a better immune response but shows more oxidative and inflammatory reactions than the montanide. Besides, saponin has site-specific adverse reactions and induces hemolytic crises. Thus, montanide can be explored as alternate adjuvant for mycoplasma vaccines.

## Institutional Review Board Statement: 

This study was approved by Institute Animal Ethics Committee (IAEC) vide Order No. AU/FVSc/PS-57/4298-99.

## Figures and Tables

**Figure 1 microorganisms-10-01634-f001:**
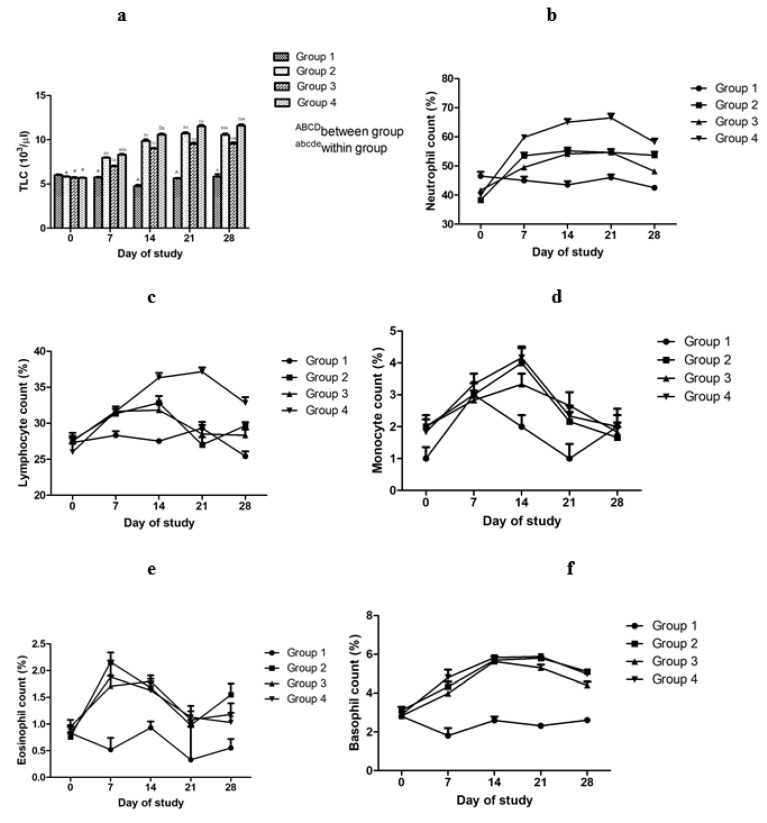
Effectof adjuvanted, sonicated antigens on leucocyte indices in rabbits. (**a**) Total leucocyte count (TLC) in 10^3^/μL. (**b**) Neutrophil count in %. (**c**) Lymphocyte count in %. (**d**) Monocyte count in %. (**e**) Basophil count in %. (**f**) Eosinophil count in %.

**Figure 2 microorganisms-10-01634-f002:**
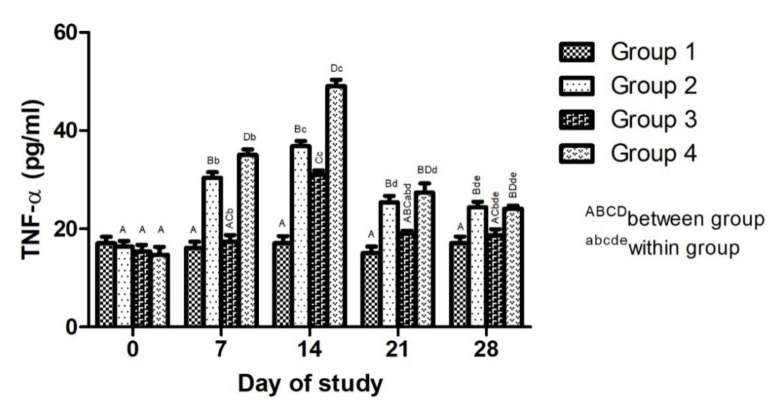
Effect of adjuvanted, sonicated antigens on levels of tumor necrosis factor alpha (TNF-α) (pg/mL) in rabbits.

**Figure 3 microorganisms-10-01634-f003:**
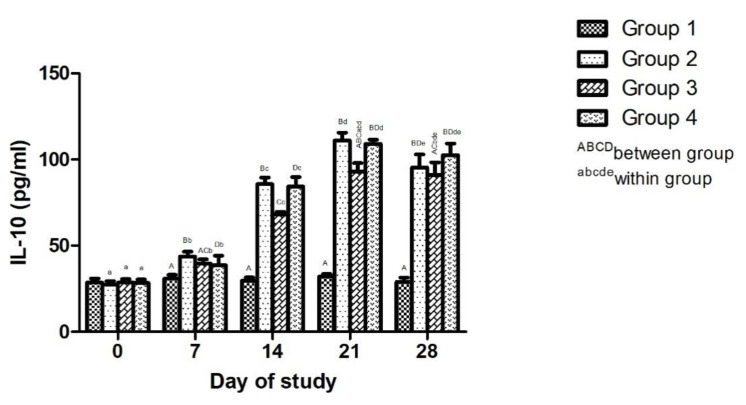
Effect of adjuvanted, sonicated antigens on levels of interleukin10 (IL-10) (pg/mL) in rabbits.

**Figure 4 microorganisms-10-01634-f004:**
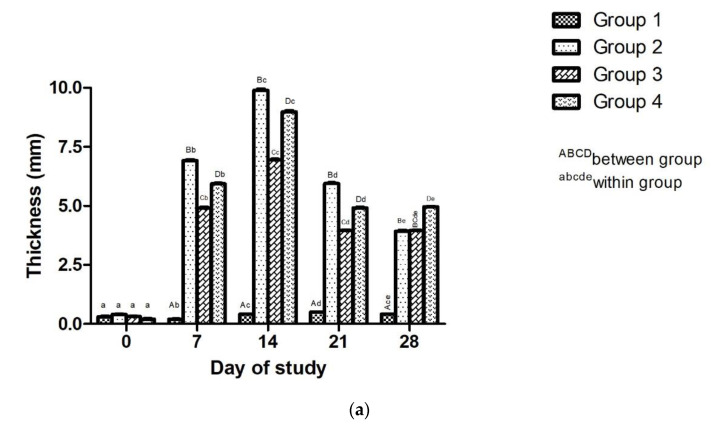
Effect of adjuvanted, sonicated antigens on hypersensitivity in rabbits. (**a**) Ear thickness (mm). (**b**) Ear hypersensitivity to antigen. (**c**) Histopathological changes in ear [Scale bar 100 μm].

**Figure 5 microorganisms-10-01634-f005:**
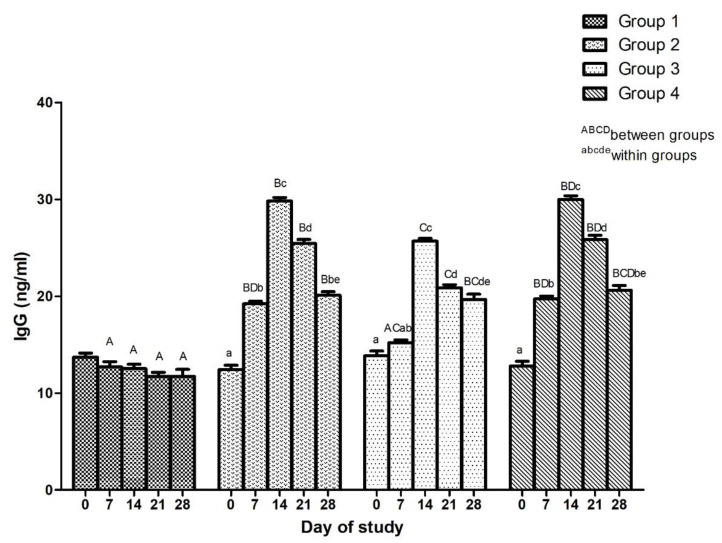
Effect of adjuvanted, sonicated antigens on immunoglobulin G (IgG) levels (ng/mL) in rabbits.

**Figure 6 microorganisms-10-01634-f006:**
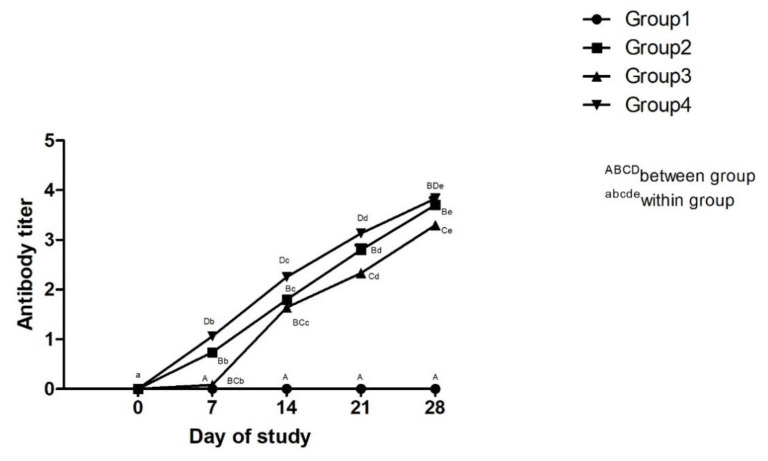
Effect of adjuvanted, sonicated antigens on antibody titer in rabbits.

**Table 1 microorganisms-10-01634-t001:** Effect of adjuvanted, sonicated antigens on TOS (μmol H_2_O_2_ equivalent/L) in rabbits.

Day of Study	Group 1	Group 2	Group 3	Group 4
0 day	3.16 ± 0.17	3.62 ± 0.13 ^a^	3.67 ± 0.14 ^a^	3.78 ± 0.02 ^a^
7 day	3.58 ± 0.34 ^A^	40.31 ± 0.71 ^Bb^	28.12 ± 0.44 ^CBb^	40.08 ± 0.65 ^BDb^
14 day	3.82 ± 0.40 ^A^	31.37 ± 0.33 ^Bc^	15.31 ± 0.55 ^Cc^	28.80 ± 0.23 ^Dc^
21 day	4.23 ± 0.35 ^A^	10.44 ± 0.33 ^Bd^	5.25 ± 0.17 ^Cd^	8.24 ± 0.19 ^Dd^
28 day	4.09 ± 0.31 ^A^	5.45 ± 0.20 ^Bae^	4.31 ± 0.10 ^ACade^	4.10 ± 0.03 ^ACDae^

^ABCD^ values with different superscript differ significantly (*p* < 0.05) between groups. ^abcde^ values with different superscript differ significantly (*p* < 0.05) within groups.

**Table 2 microorganisms-10-01634-t002:** Effect of adjuvanted, sonicated antigens on TAS (μmol trolox equivalent/L) in rabbits.

Day of Study	Group 1	Group 2	Group 3	Group 4
0 day	0.59 ± 0.07	0.4589 ± 0.08 ^a^	0.44 ± 0.09 ^a^	0.36 ± 0.05 ^a^
7 day	0.42 ± 0.07 ^A^	0.12 ± 0.00 ^Bb^	0.27 ± 0.01 ^ACab^	0.13 ± 0.01 ^BCDb^
14 day	0.59 ± 0.07 ^A^	0.1376 ± 0.01 ^Bbc^	0.2485 ± 0.01 ^BCbc^	0.12 ± 0.00 ^BCDbc^
21 day	0.67 ± 0.07 ^A^	0.13 ± 0.01 ^Bbcd^	0.24 ± 0.01 ^BCbcd^	0.21 ± 0.02 ^BCDbcd^
28 day	0.59±0.07 ^A^	0.24±0.00 ^Bbcde^	0.32±0.00 ^BCabcde^	0.33±0.01 ^BCDae^

^ABCD^ values with different superscript differ significantly (*p* < 0.05) between groups. ^abcde^ values with different superscript differ significantly (*p* < 0.05) within groups.

**Table 3 microorganisms-10-01634-t003:** Effect of adjuvanted, sonicated antigens on total erythrocyte count (million cells/cmm) in rabbits.

Day of Study	Group 1	Group 2	Group 3	Group 4
0 day	6.13 ± 6.13	5.88 ± 0.12 ^a^	6.53 ± 0.33 ^a^	5.91 ± 0.33 ^a^
7 day	6.11 ± 0.41 ^A^	4.43 ± 0.13 ^Bb^	5.68 ± 0.10 ^ACb^	4.68 ± 0.11 ^BDb^
14 day	5.98 ± 0.37 ^A^	4.96 ± 0.06 ^Bc^	5.35 ± 0.12 ^ABCbc^	5.20 ± 0.10 ^BCDc^
21 day	6.08 ± 0.28 ^A^	5.83 ± 0.08 ^ABad^	6.25 ± 0.11 ^ABCad^	5.78 ± 0.07 ^ABDad^
28 day	6.13 ± 0.33 ^A^	6.43 ± 0.12 ^ABe^	6.85 ± 0.09 ^BCae^	6.48 ± 0.11 ^ABCDe^

^ABCD^ values with different superscript differ significantly (*p* < 0.05) between groups. ^abcde^ values with different superscript differ significantly (*p* < 0.05) within groups.

**Table 4 microorganisms-10-01634-t004:** Effect of adjuvanted, sonicated antigens on Hb (g/dL) in rabbits.

Day of Study	Group 1	Group 2	Group 3	Group 4
day 0	12.50 ± 0.67	13.00 ± 0.57 ^a^	13.00 ± 0.96 ^a^	12.83 ± 0.79 ^a^
day 7	11.50 ± 0.84	9.25 ± 0.21 ^b^	10.50 ± 0.42 ^b^	9.66 ± 0.35 ^b^
day 14	11.66 ± 0.66 ^A^	9.00 ± 0.22 ^Bbc^	10.41 ± 0.52 ^ABCbc^	9.16 ± 0.21 ^BCDbc^
day 21	12.50 ± 0.76 ^A^	9.33 ± 0.21 ^Bbcd^	11.16 ± 0.54 ^ABCabcd^	9.66 ± 0.33 ^BCDbcd^
day 28	12.66 ± 0.88	10.66 ± 0.33 ^bde^	12.50 ± 0.34 ^abcde^	10.58 ± 0.37 ^bcde^

^ABCD^ values with different superscript differ significantly (*p* < 0.05) between groups. ^abcde^ values with different superscript differ significantly (*p* < 0.05) within groups.

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
