# Peer review of "Immune and Oxidative Response against Sonicated Antigen of Mycoplasma capricolum subspecies capripneumonia—A Causative Agent of Contagious Caprine Pleuropneumonia"

_microorganisms, 2022, doi:10.3390/microorganisms10081634_

Round 1

Reviewer 1 Report

The Authors have evaluated the immune response and the oxidative stress following administering a vaccine against Mycoplasma capricolum subspecies. capripneumonia in male angora Rabbits, divided into 4 groups in accordance study's aims. Although the research interests the reader, a significant revision is required to consider the manuscript for publication.

The introduction must be improved, involving information about oxidative stress in course of pulmonary infection and veterinary medicine. Similarly, the consideration above applies to the immune evaluation.

Materials and methods are more detailed. The evaluation of the time of oxidative stress is not clear. 

Discussion must be improved and better related to the results, which must be commentated. 

A significant English revision is necessary with a native speaker. The sentence is often very long, very confusing, and difficult to understand. Often the paragraph are not correlated between their. 

Author Response

Respected Sir 

The manuscript has been revised as per your kind suggestions. Below is the point-by-point response to your comments:

Reviewer 1:

  1. The introduction must be improved, involving information about oxidative stress in course of pulmonary infection and veterinary medicine. Similarly, the consideration above applies to the immune evaluation.

Reply: Introduction has been revised and information about oxidative stress and immune response in course of pulmonary infection and veterinary medicine has been added.

  1. Materials and methods are more detailed. The evaluation of the time of oxidative stress is not clear. 

Reply: Materials and methods section has been revised and evaluation of the time period of oxidative stress has been added (0, 7, 14, 21, 28 day)

  1. Discussion must be improved and better related to the results, which must be commentated. 

Reply: Discussion has been revised and related to results with addition and dleletion of some parts.

  1. A significant English revision is necessary with a native speaker. The sentence is often very long, very confusing, and difficult to understand. Often the paragraph are not correlated between their. 

Reply: Englsih revision by native English speaker has been done. Some sentences have been shortened or deleted. Paragraphs have been correlated by revising sentences.

Hope the manuscript has improved now.

Thank You

With Regards

Reviewer 2 Report

The article under consideration is devoted to the study of the method of immune response in contagious goat pleuropneumonia. In my opinion, the article is well written, the data obtained are substantiated, the conclusions correspond to the experimental data. In principle, the article can be published, but I have a non-essential request: on some drawings, the inscriptions cannot be read. For example: Fig. 1 is not readable, I understood what is written there based on the text of the article. Fig. 3-4. The inscriptions are illegible.

Author Response

Respected Sir

All your kind comments have been considered and manuscript revised as suggested. Below is the point-by-point response to your comments:

  1. The article under consideration is devoted to the study of the method of immune response in contagious goat pleuropneumonia. In my opinion, the article is well written, the data obtained are substantiated, the conclusions correspond to the experimental data. In principle, the article can be published, but I have a non-essential request: on some drawings, the inscriptions cannot be read. For example: Fig. 1 is not readable, I understood what is written there based on the text of the article. Fig. 3-4. The inscriptions are illegible.

Reply: Thank You Sir for appreciating the article. Inscriptions on drawings are visible when the images are enlarged as in revised file. Fig. 1 is revised and is now readable. Fig. 3-4 inscriptions have been revised and are now legible. All the comments have been considered and figures revised accordingly as in the revised manuscript.

Hope the manuscript has improved now.

Thank You

With Regards

Round 2

Reviewer 1 Report

Well done. Thank you for having accepted my suggestion.